# Research Tendency and Frontiers of Multifocal Lenses in Myopic Control in the Past Two Decades: A Bibliometric Analysis

**DOI:** 10.3390/healthcare13020204

**Published:** 2025-01-20

**Authors:** Lingli Jiang, Manrong Yu, Jiangxiong Cai, Yingying Wang, Hao Hu, Minjie Chen

**Affiliations:** 1Department of Ophthalmology, The Wenling Hospital, Wenzhou Medical University, Wenling 317500, China; wlyyjll@tzc.edu.cn (L.J.); wlyycjxll@tzc.edu.cn (J.C.); wlyywyy@tzc.edu.cn (Y.W.); wlyyhuhao@tzc.edu.cn (H.H.); 2Eye Institute and Department of Ophthalmology, Eye & ENT Hospital, Fudan University, Shanghai 200031, China; 16111260009@fudan.edu.cn; 3NHC Key Laboratory of Myopia and Related Eye Diseases, Key Laboratory of Myopia and Related Eye Diseases, Chinese Academy of Medical Sciences, Shanghai 200031, China; 4Shanghai Key Laboratory of Visual Impairment and Restoration, Shanghai 200031, China

**Keywords:** multifocal lenses, myopic control, contact lens, children, bibliometric, VOSviewer, CiteSpace

## Abstract

**Background:** This study aimed to analyze the research progress on the use of a multifocal lens for myopic control throughout the 21st century, utilizing bibliometric analysis. **Methods:** Publications related to multifocal lenses from 2001 to 2024 were searched on the Web of Science core collection (WoSCC) database. VOSviewer (Version 1.6.19) and Bibliometrix package (Version 4.3.0) were used to perform the bibliometric analysis. Primary information including the publication number per year, country or region, journal, keyword, and title of the literature was extracted and analyzed. **Results:** A total of 234 articles from 31 countries were retrieved. The number of publications related to multifocal lenses had a rapid growth phase from 2016 to 2024. The United States, Australia, China, and Spain emerged as leading contributors to the research landscape. *Ophthalmic and Physiological Optics* was the most popular journal in this field, and the most frequently cited article was published in *Optometry and Vision Science*. Myopic progression was the main topic in this research field as well as the principal keywords of emerging research hotspots. **Conclusions:** Our bibliometric study provides a pioneering overview of the research trends and evolution in the application of a multifocal lens for myopic control. These findings provide a deep insight into current research frontiers and hot directions, offering a valuable reference for further research.

## 1. Background

Myopia is the most common refractive error and has become a widespread global health challenge in recent decades [1]. There has been a gradual increase in the pooled prevalence of myopia, ranging from 24.32% in 1990 to 35.81% in 2023, with projections indicating that this prevalence is expected to reach 39.80% in 2050 [2]. The potential etiology of myopia is increasingly recognized as multifactorial, encompassing a complex interplay of environmental, lifestyle, educational, and genetic factors [3,4]. High myopia, especially pathological myopia, can lead to a variety of ocular complications and, in severe cases, result in blindness [5]. In response to these concerns, myopic control has become a critical issue in public health in recent years, aiming to reduce the associated risks of vision-impairing diseases [6,7].

Beyond behavioral interventions, a variety of treatment strategies, including optical and pharmacological options, have been explored to control myopic progression [8]. Multifocal lenses, an optical intervention strategy, encompass both multifocal contact lenses (MCLs) and multifocal spectacle lenses, which have been widely used to mitigate myopic progression. These innovative lens designs have been engineered to induce a peripheral defocus on the retina. This aims to reduce peripheral hyperopic defocus, thereby reducing a key stimulant for axial elongation [9].

Scientometric analysis is a mathematical and statistical method that facilitates a comprehensive synthesis of the scientific literature [10]. Various software tools have been crafted to aid researchers in their exploration of the scientific literature, such as CiteSpace and VOSviewer. This analytical approach not only enables systematic retrospective assessments but also serves as a powerful instrument for predicting the trajectory of future research trends. This study aimed to employ bibliometric techniques to analyze the evolution of research pertaining to multifocal products as a means of myopic control, spanning a 24-year period.

## 2. Methods

### 2.1. Literature Resources

The terms, “multifocal” and “refractive error or myopia or nearsightedness” were used as topics, excluding the topics “intraocular”, ”electroretinogram”,” choroidal neovascularization” and “VEP” in the Web of Science core collection databases, between 1 January 2001 and 26 October 2024, with no language restriction. The literature search was concluded on 26 October 2024. The methodical process of the scientometric research is shown in Figure 1. The articles and reviews were refined and used for scientometric analysis.

### 2.2. Data Analysis and Visualization

The raw data from the literature search were systematically analyzed using the Bibliometrix package (Version 4.3.0, Aria, M., & Cuccurullo, C., 1 July 2024) within the R programming environment (Version 4.2.2, R Core Team, 2022). Primary information, including the annual publication counts, geographical distribution of contributing countries or regions, associated journals, prevalent keywords, and titles of the literature, were extracted using Bibliometrix. VOSviewer (Version 1.6.19, Leiden University, Leiden, The Netherlands) was used to optimize and supplement the analytical process, particularly in refining the graphical representations of the data. Visualization of the extracted information was performed using a combination of tools, including the R software (Version 4.4.2) for dynamic data manipulation and Origin software (Version 2022, OriginLab Corporation, Northampton, MA, USA) for its sophisticated graphing capabilities and for creating interactive and publication-quality figures.

## 3. Results

### 3.1. General Analysis

A total of 234 published research articles, comprising 196 original research articles and 38 review papers, published from 1 January 2001 to 26 October 2024, were included in this study. The trajectory of publication growth in this subject area is characterized by two distinct phases: an initial slow phase spanning the years 2001 to 2015, followed by a period of accelerated expansion from 2016 to 2024 (refer to Figure 2). The rapid growth in research interest and output during the latter period is likely related to heightened focus on myopic control as a critical issue in public health and the emergence of innovative approaches for controlling the progression of myopia, such as the use of low-dose atropine eye drops and the advent of multifocal products.

### 3.2. National/Regional Analysis

A geographical analysis of the contributions to multifocal research revealed a broad international engagement, with a total of 31 countries and regions participating in the scholarly discourse. When utilizing the Bibliometrix package for bibliometric analysis, the national affiliation of a paper is commonly determined based on the country information of the corresponding author. A single-country publication (SCP) is a publication whose authors are all from the same country, while a multiple-country publication (MCP) has authors from multiple countries, also known as a joint publication. As shown in Figure 3A, the United States led the publication count of 51 SCPs and 11 MCPs, followed by China with 20 SCPs and 10 MCPs, Australia with 21 SCPs and 9 MCPs, and Spain with 10 SCPs and 8 MCPs. Also, the United States has the highest proportion (82.3%) of independent publications among countries with more than five publications. The proportions of MCPs for most countries ranged from 30 to 50%, including Australia, China, Spain, and Germany. The partnership among different regions highlighted the collaborative nature of this research field. Figure 3B illustrates the strong collaborative ties, particularly the significant joint efforts between USA and England, Australia, and China.

### 3.3. Journal Analysis

All publications were published in 79 types of journals. The top 10 journals are shown in Figure 4. *Ophthalmic and Physiological Optics* emerged as the preeminent journal in this field, featuring the highest number of articles dedicated to this topic. Successively, *Contact Lens Anterior Eye* ranked as the second most prolific publisher in this area, followed by, Optometry and Vision Science, *Eye Contact Lens Science and Clinical Practice*, and *British Journal of Ophthalmology*, which consistently contributed to the advancement of knowledge on this topic.

### 3.4. Literature and Citation References

The ten most frequently cited articles pertinent to this research domain are shown in Table 1. The article gathering the most citations reported the impact of soft multifocal contact lens wear on the progression of axial myopia in children [11]. The second most cited article primarily outlines various strategies for the prevention of myopia and its progression [12]. Subsequent highly cited works examined the performance of high-add-power multifocal contact lenses in the context of myopia management and the alteration in myopia progression and axial elongation through peripheral defocus [13,14]. The top 10 co-cited references about this research are shown in Table 2. These articles reported significant findings for researchers to explore the effect of multifocal lenses on myopia control.

### 3.5. Keyword Analysis

According to the principles of bibliometrics, keywords represent the focal points and prevailing trends within a given research field [15]. In this analysis, the visualization of keywords is depicted through a heatmap (Figure 5A) and a co-occurrence network diagram (Figure 5B). As shown in Figure 5A, the heatmap illustrates an increase in keyword frequency by the year, with “myopia” emerging as the most prominent topic, followed by other significant keywords such as “myopia control”, “orthokeratology”, “contact lenses”, and “multifocal contact lenses”.

A co-occurrence network presents a more nuanced perspective by grouping keywords into distinct clusters, color-coded according to their correlations. As observed in Figure 5B, the keyword “myopia” appeared as the largest node within the network, aligning with the central research topic of this study.

## 4. Discussion

The prevalence of myopia is on an alarming upward trend, with projections estimating that over five billion people could be affected by 2050 [16]. The increasing early onset of myopia, coupled with the high incidence of high myopia, requires effective strategies for management [4,17,18]. Remarkable improvements have been evidenced in optical and pharmacological strategies to slow the progression of myopia in the past decades [19]. Among these developments, multifocal lenses have attracted considerable attention for their potential in myopic control. This category includes MCLs and multifocal spectacle lenses, which have emerged as particularly compelling options due to their innovative design and demonstrated efficacy in managing myopia.

### 4.1. Multifocal Lens Designs and Types

The design of MCLs for myopia control typically features a central zone dedicated to distance correction, surrounded by concentric peripheral zones that incorporate a relative addition power of +2.00 diopters (D) or greater [20]. The extended depth of focus (EDOF) lens is an innovative design that leverages a graduated distribution of power from the central to peripheral regions. This design introduces higher-order aberrations to refine the retinal image quality, with powers up to +1.75 or +2.50 D, which can optimize focus on points at and in front of the retina, while intentionally degrading the focus for points behind the retina [21]. Center-distance MCLs induce positive spherical aberration, while center-near MCLs cause negative spherical aberration [22]. MCLs are designed to create varying levels of peripheral myopic defocus on the retina. It has been observed that individuals with lower myopia may experience more myopic defocus with MCLs, which can be an important consideration when tailoring treatment strategies for myopic control [23].

Additionally, various multifocal spectacles lens designs have been applied. The MyoVision (Carl Zeiss Vision, Aalen, Germany) spectacle lenses feature an asymmetric design, with a clear central aperture that spans approximately 10 mm on either side of the central horizontal meridian and an equivalent distance below, ensuring sharp vision for tasks requiring convergence and a downward gaze [24]. Furthermore, the Defocus Incorporated Multiple Segment (DIMS) spectacle lenses (MiYOSMART, HOYA Corporation, Tokyo, Japan) incorporate a zonal configuration composed of miniature and circular lenslets, each approximately 1 mm in diameter. These lenslets are designed to provide additional power, but images from each individual lenslet do not converge to create a single image in the focal plane corresponding to the add power, but rather multiple separate images [25].

### 4.2. Multifocal Lenses and Myopic Control

Noteworthy outcomes have been documented in prior research regarding the efficacy of multifocal lenses in managing myopic progression [26,27,28,29,30,31,32]. In a 1-year prospective, randomized, contralateral, cross-over clinical trial, EDOF (Pegavision, pegavision.com, Taoyuan, Taiwan) and MiSight^®^ (CooperVision, San Ramon, CA, USA) lenses showed comparable efficacy in slowing myopia, with no observed rebound effect when transitioning from EDOF or MiSight^®^ lenses to single-vision lenses (SVLs) [26]. The change in spherical equivalent (SE) refractive error and axial length (AL) was significantly lower for EDOF and MiSight^®^ CLs than for contralateral single-vision contact lenses in 68% to 94% of participants [26]. Another study revealed that the weighted mean differences in SEs between multifocal lenses and SVLs were 0.29 D, 0.46 D, and 0.64 D, while the weighted mean differences in ALs were −0.12 mm, −0.19 mm, and −0.26 mm at the first, second, and third years, respectively [27].

The results from a 3-year follow-up study showed that the efficacy of myopia control was maintained throughout the third year in children who had used DIMS spectacles during the preceding two years. This persistent effect was also observed in children who transitioned from SVLs to DIMS lenses [28]. Highly aspherical lenses (HALs, Essilor Stellest, Essilor International, Charenton-le-Pont, France) are aspherical lenses that create a zone of myopic defocus in front of the retina, achieved through 11 concentric rings of contiguous lenslets [29]. The study showed that HALs effectively decelerated myopia progression, with no rebound observed when patients switched from HALs to SVLs [30]. Furthermore, a recent study found that Chinese children wearing HALs experienced reduced myopic progression and axial elongation compared to those using DIMS lenses, highlighting the potential of HALs as an effective myopia management option [31].

Another novel design of spectacle lenses was also reported. The Cylindrical Annular Refractive Element (CARE, Carl Zeiss Vision International GmbH, Aalen, Germany) spectacle lenses feature a central clear aperture of 9.4 mm in diameter, which ensures effective and consistent visual correction, complemented by a peripheral side-vision zone encompassing an array of annular microcylinders. In a 2-year clinical trial conducted with Chinese children, the interim results from the first year showed that CARE spectacle lenses significantly slowed myopia progression [32]. Further research is warranted to substantiate the efficacy of multifocal spectacle lenses across various racial and ethnic groups, as well as to evaluate any potential rebound effects associated with their use.

In recent years, AL has become an equally important parameter for evaluating myopia progression as SE, especially in studies evaluating the efficacy of contact lens in myopia control [33,34,35]. The development of Optical Coherence Tomography (OCT) has revolutionized the field by enabling detailed choroidal imaging and quantitative analysis [36]. A range of myopia management strategies, including atropine eye drops [37], Orthokeratology (Ortho-K) [38], repeated low-level red light therapy [39] and other treatments that induce myopic defocus, have been shown to influence choroidal thickness and slow down axial elongation [40,41,42]. More studies should be designed to validate whether choroidal thickness could serve as a predictor for the efficacy of myopia interventions.

### 4.3. Multifocal Lens and Visual Quality

Visual acuity (VA) and contrast sensitivity represent key components of spatial vision, routinely assessed in both clinical practice and research settings. Previous clinical studies have reported a spectrum of visual compromises in young adults wearing MCLs. Although VA remains sufficiently high for the majority of these patients, there persists a notable prevalence of complaints regarding diminished VA and dysphotopsia [43]. Various studies have consistently reported normal high-contrast visual acuity in individuals wearing MCL, dual-focus (DF) CL, HAL, Ortho-K, and DIMS spectacle lenses [9,44,45,46,47,48,49,50]. However, impaired low-contrast visual acuity has also been noted in these populations. Comparative studies revealed a decrement in contrast sensitivity when wearing MCL or DF contact lenses as opposed to single-vision lenses [44,51]. This trend was corroborated under both photopic and mesopic lighting conditions in another investigation [52]. But in a double-masked study, contrast sensitivity function was reduced specifically with +2.0 D add power MCLs, but not with +4.0 D add power lenses [53]. Another study showed that lenses with a medium sized central inner zone yielded superior photopic contrast sensitivity, particularly at high spatial frequencies [46].

An increase in higher-order aberrations (HOAs) may generate a stimulus akin to visual deprivation, which can degrade retinal image quality and elicit visual signals that encourage ocular growth. This is postulated because the wavefront alterations induced during near work could mimic the effects of hyperopic defocus [54]. One study showed that the design of DF CLs with a smaller central distance diameter of 2.1 mm induced a lower degree of HOAs for a 5 mm pupil diameter, in contrast to those with a larger central distance diameter of 4 mm [46]. Meanwhile, another investigation found that a design with a central diameter of 3.36 mm tailored for distance vision increased values of HOAs compared with SVLs [51]. Similar results were found for MCLs with a center distance diameter of 3 mm, highlighting the relationship between multifocal lens configuration and the induction of HOAs [55].

The role of accommodation in myopia development and progression is still unclear. Accommodative lag is theorized to be linked to the onset and development of myopia, because it yields hyperopic defocus on the retina [43]. Lam et al. reported that after wearing DIMS lenses for 2 years, there was a significant reduction in accommodative lag [56]. Similar changes were also noted after wearing SV lenses in these Chinese myopic children [47]. Therefore, these findings suggest that the mechanism by which DIMS lenses decelerate myopia progression is unlikely due to alterations in accommodation. Kate et al. also found a similar result in young myopic adults using MiSight lenses, further supporting the notion that the impact of these lenses on myopia control extends beyond mere accommodation modulation [57].

Although the United States, Australia, China, and Spain emerged as leading contributors to the research landscape, the majority of the top 10 most cited articles originated from the United States who also led the highest proportion of independent publication, indicating the leading position of the United States in the field of myopia research. Hopefully, in the future, scholars around the world will endeavor to improve the quality of their research and publishing to expand their academic impact. Since these studies reflect regional contributions to myopia research and inter-regional academic collaborations, they are expected to provide useful information for national health policymakers and researchers in related fields. In addition, due to different ethnic groups and national policies, the prevention and management of myopia are different from country to country. In the future, multi-regional communications and collaborations are necessary.

The current study is the first bibliometric analysis focused on the application of multifocal lenses in myopic control, utilizing the literature spanning the entire 21st century. Nevertheless, there are still several limitations inherent in our study. Data extraction was confined to the Web of Science database, thereby excluding other significant medical databases such as PubMed and Scopus. We filtered studies published in English, which may mean non-English writing papers were underestimated. Also, only articles and reviews were retrieved, whereas meeting abstracts, editorial materials, letters, and proceeding papers were excluded in our analysis, which also may have resulted in incomplete data collection. In addition, there was a delay between the completion of this article and its inclusion in databases, which could have affected the timeliness of the analysis.

## 5. Conclusions

In conclusion, this study offers a comprehensive and objective analysis of the principal studies concerning the use of multifocal products for myopia control. Our findings highlight the research hotspots, key research directions, and leading countries and regions over the past 24 years. In addition, using the Web of Science database in conjunction with VOSviewer software enabled a detailed visualization, analysis, and identification of the most influential articles and their interconnecting citation networks to date. This review concluded with the prevalence, mechanisms, progression, comparative analyses, and various models of multifocal products in the current literature, as derived from the most frequently occurring keywords in our keyword analysis.

## Figures and Tables

**Figure 1 healthcare-13-00204-f001:**
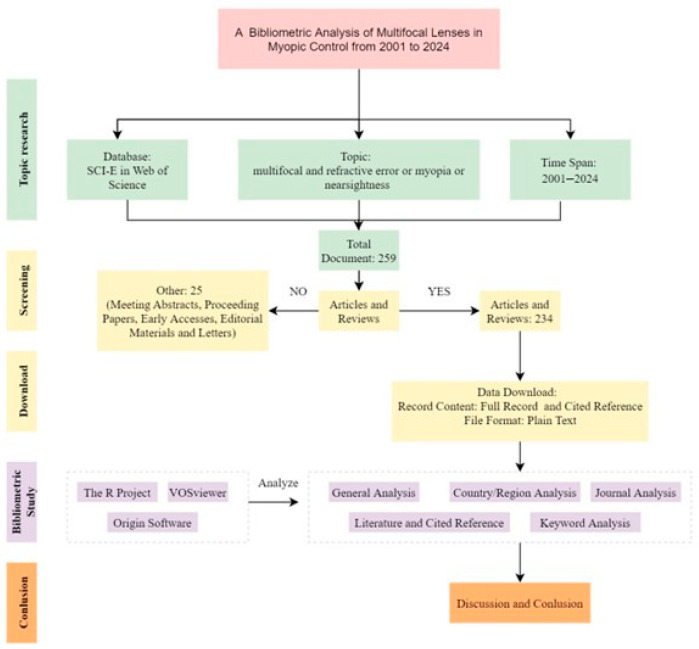
Flow chart of scientometric analysis.

**Figure 2 healthcare-13-00204-f002:**
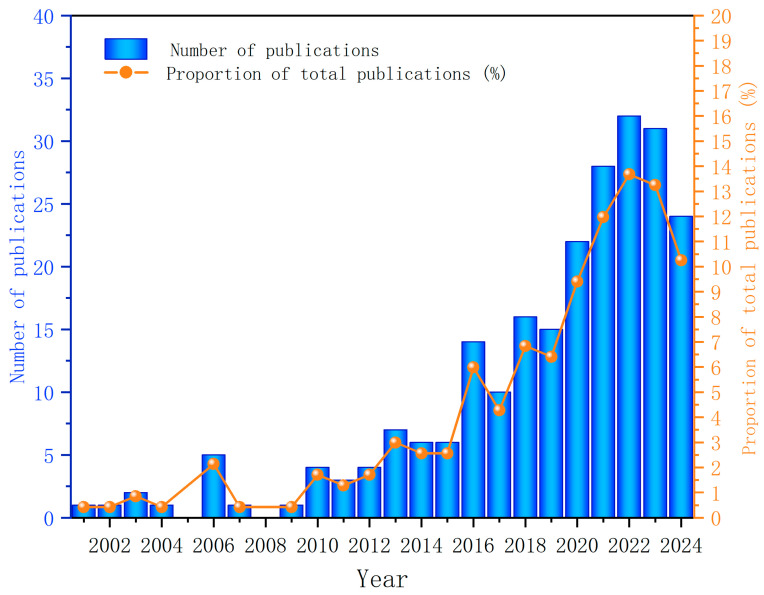
Annual publications between 2001 and 2024.

**Figure 3 healthcare-13-00204-f003:**
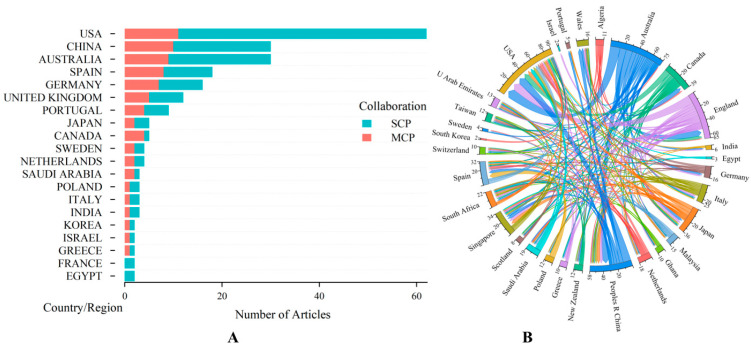
(**A**) Number of publications by country or region. SCP, single-country publication, a publication in which all authors are from the same country; MCP, multiple-country publication, a publication with authors from multiple countries, known as a joint publication. (**B**) The cooperative relationships between a country or region, where the wider the band, the stronger the cooperation between the two regions.

**Figure 4 healthcare-13-00204-f004:**
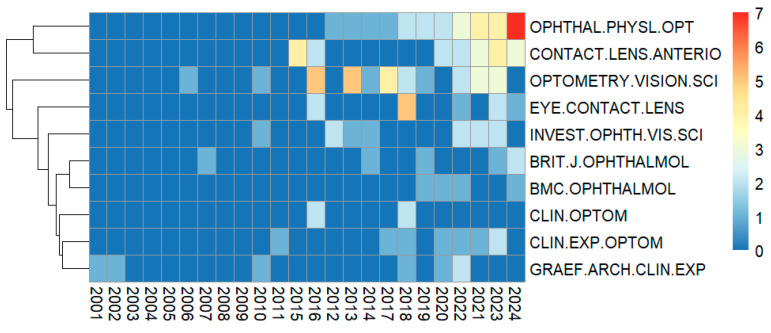
Number of publications by journal, with the sidebar using a color-coded system to denote the corresponding counts. The left dendrogram on this heatmap illustrates hierarchical clustering outcomes, organizing journals into a tree structure based on similarity. Vertical lines mark journal cluster merges, with height indicating dissimilarity; greater heights suggest higher dissimilarity. Horizontal lines show journal positions, each linking to a tree node. The tree structure reveals hierarchical journal relationships, from individual terms to the overall cluster.

**Figure 5 healthcare-13-00204-f005:**
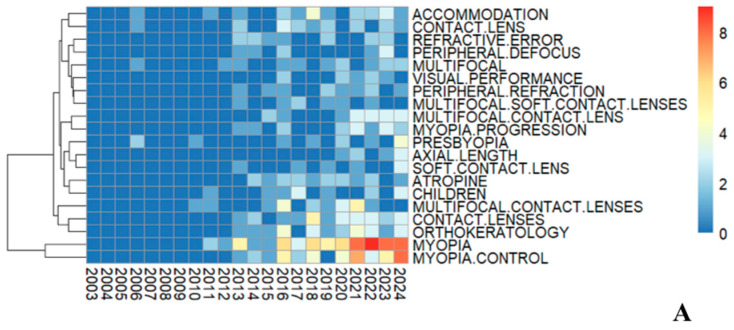
Keyword analysis. (**A**) Annual frequency of keywords, with the sidebar using a color-coded system to denote the corresponding counts. The left dendrogram on this heatmap illustrates hierarchical clustering outcomes, organizing keywords into a tree structure based on similarity. Vertical lines mark keyword cluster merges, with height indicating dissimilarity; greater heights suggest higher dissimilarity. Horizontal lines show keyword positions, each linking to a tree node. The tree structure reveals keyword hierarchical relationships, from individual terms to the overall cluster. (**B**) Keyword co-occurrence network. The size and coloration of the nodes are representative of the volume of keywords and the clusters they form. The interconnecting lines, varied in color, illustrate instances where pairs of keywords were published within the same article.

**Table 1 healthcare-13-00204-t001:** The top 10 most cited articles related to multifocal lenses on myopia control.

Ranking	Cited Number	Title of Article	Year	Journal			
				Name	Country/ Region	Impact Factor (2023)	H-Index (2023)
1	195	Multifocal Contact Lens Myopia Control	2013	Optometry and Vision Science	USA	1.6	115
2	170	IMI Prevention of Myopia and Its Progression	2021	Investigative Ophthalmology and Visual Science	USA	5	252
3	159	Effect of High Add Power, Medium Add Power, or Single-Vision Contact Lenses on Myopia Progression in Children The BLINK Randomized Clinical Trial	2020	JAMA—Journal of the American Medical Association	USA	63.5	768
4	153	Axial Eye Growth and Refractive Error Development Can Be Modified by Exposing the Peripheral Retina to Relative Myopic or Hyperopic Defocus	2014	Investigative Ophthalmology and Visual Science	USA	5	252
5	142	Myopia Control with Bifocal Contact Lenses: A Randomized Clinical Trial	2016	Optometry and Vision Science	USA	1.6	115
6	139	A Review of Current Concepts of the Etiology and Treatment of Myopia	2018	Eye and Contact Lens	USA	2	67
7	104	The Effects of Age, Refractive Status, and Luminance on Pupil Size	2016	Optometry and Vision Science	USA	1.6	115
8	95	Myopia Control: A Review	2016	Eye and Contact Lens	USA	2	67
9	91	Studies using concentric ring bifocal and peripheral add multifocal contact lenses to slow myopia progression in school-aged children: a meta-analysis	2017	Ophthalmic and Physiological Optics	England	2.8	82
10	90	Comparison of multifocal and monovision soft contact lens corrections in patients with low-astigmatic presbyopia	2006	Optometry and Vision Science	USA	1.6	115

**Table 2 healthcare-13-00204-t002:** The top 10 co-cited references about this research.

Ranking	Cited Number	Title of Article	Year	Journal			
				Name	Country /Region	Impact Factor (2023)	H-Index (2023)
1	103	Effect of dual-focus soft contact lens wear on axial myopia progression in children	2011	Ophthalmology	USA	13.1	285
2	81	Decrease in rate of myopia progression with a contact lens designed to reduce relative peripheral hyperopia: one-year results	2011	Investigative Ophthalmology and Visual Science	USA	5	252
3	75	Multifocal contact lens myopia control	2013	Optometry and Vision Science	USA	1.6	115
4	71	Defocus Incorporated Soft Contact (DISC) lens slows myopia progression in Hong Kong Chinese schoolchildren: a 2-year randomised clinical trial	2014	British Journal of Ophthalmology	England	3.7	179
5	70	Global Prevalence of Myopia and High Myopia and Temporal Trends from 2000 through 2050	2016	Ophthalmology	USA	13.1	285
6	61	A 3-year Randomized Clinical Trial of MiSight Lenses for Myopia Control	2019	Optometry and Vision Science	USA	1.6	115
7	59	Retardation of myopia in Orthokeratology (ROMIO) study: a 2-year randomized clinical trial	2012	Investigative Ophthalmology and Visual Science	USA	5	252
8	54	Myopia Control with Bifocal Contact Lenses: A Randomized Clinical Trial	2016	Optometry and Vision Science	USA	1.6	115
9	51	Relative peripheral hyperopic defocus alters central refractive development in infant monkeys	2009	Vision Research	England	1.5	184
10	50	Effect of High Add Power, Medium Add Power, or Single-Vision Contact Lenses on Myopia Progression in Children: The BLINK Randomized Clinical Trial	2020	JAMA—Journal of the American Medical Association	USA	63.5	768

## Data Availability

The datasets used and analyzed during the current study are available from the corresponding author on reasonable request.

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
