# Peer review of "Research Tendency and Frontiers of Multifocal Lenses in Myopic Control in the Past Two Decades: A Bibliometric Analysis"

_healthcare, 2025, doi:10.3390/healthcare13020204_

Round 1
Reviewer 1 Report
Comments and Suggestions for Authors
The manuscript is interesting and deals with important topic, however, it needs revision.
1. The prevalence of myopia should ba added in the Introduction section.
2. Did the Authors use PRISMA flowchart to select the articles. If no, they should explain why
3. Risk of bias should be assessed (selection, performance, detection, attrition, reporting bias) in the Methods section DOI: 10.3390/jcm13154400
4. Too few limitations of the study were listed. This paragraph should be expanded
5. Clinical value of the study should be highlighted
Additional comments:
1. An explanation should be added as to what distinguishes this manuscript from others.
2. Do the Authors plan further studies in this field?
Author Response
Dear reviewer 1,
Thank you so much for giving us this opportunity to revise our manuscript, which was significantly improved with your constructive suggestions. The corresponding revisions highlighted in the re-submitted files, and the point-to-point responses to the reviewer’s comments are listed below:
Comments:
The manuscript is interesting and deals with important topic, however, it needs revision.
- The prevalence of myopia should be added in the Introduction section.
Response: Thank you very much for your useful suggestion, the prevalence of myopia is added in the Introduction section from line 33 to line 35: “There is a gradual increase in pooled prevalence of myopia, ranging from 24.32% in 1990 to 35.81% in 2023, and projections indicate that this prevalence is expected to reach 39.80% in 2050.”
- Did the Authors use PRISMA flowchart to select the articles. If no, they should explain why
Response: Thank you for pointing out this issue. We did not use PRISMA flowchart in our study, because our study is a bibliometric analysis. Bibliometric analysis quantitatively maps research trends and impact within a field, focusing on the broader scholarly landscape without the need for a PRISMA flowchart. It examines patterns over time, identifying key papers, authors, and institutions. provides an overview of research dynamics, while PRISMA flowcharts are specific to the systematic review methodology. So the two approaches serve different purposes in academic research.
- Risk of bias should be assessed (selection, performance, detection, attrition, reporting bias) in the Methods section. DOI: 10.3390/jcm13154400
Response: Thank you very much for pointing out this issue, which is very important for us to make a deep consideration of the methodology used in this study. Unlike systematic reviews and meta-analyses, the above-mentioned characteristics of bibliometric analysis make it not typically involve the evaluation of individual study methodologies or the risk of bias within those studies. However, it is still important to consider potential biases that may arise from the databases used, search strategy, or the inclusion/exclusion criteria. So in our study, we used Web of Science core collection database, which covered most of high-quality research articles, we also used neutral terms in searching to avoid potential bias. In addition, the flow chart of scientometric analysis shows process of inclusion/exclusion of the articles. There could be potential selection bias in this study, as we mentioned in the limitation section: “The data extraction was confined to the Web of Science database, thereby excluding other significant medical databases such as PubMed and Scopus. We filtered studies published in English, which may mean non-English writing papers were underestimated. Also, only articles and reviews were retrieved, whereas meeting abstracts, editorial materials, letters, or proceedings papers were excluded in our analysis, which also might result in incomplete data collection”.
- Too few limitations of the study were listed. This paragraph should be expanded
Response: Thank you very much for your advice, which is very important in improving the quality of our manuscript. We've added two other limitations to expand this paragraph. The other two limitations are “We filtered studies published in English, which may mean non-English writing papers were underestimated. Also, only articles and reviews were retrieved, whereas meeting abstracts, editorial materials, letters, or proceedings papers were excluded in our analysis, which also might result in incomplete data collection.” You can find the corresponding sentences from line 292 to line 295.
- Clinical value of the study should be highlighted
Response: Thank you very much for your useful suggestion. As we mentioned in the conclusion, this study provides a detailed report and pioneering overview of the research trends and evolution on multifocal lens for myopic control. These findings provide a deep insight into current research frontiers and hot directions, offering a valuable reference for further research.
In addition, as you suggested, we added a paragraph to highlight the Clinical value of the study from line 277 to 287: “Although the United States, Australia, China, and Spain emerged as leading contributors to the research landscape, the majority of the top 10 most cited articles originated from the United States who also led the highest proportion of independent publication, indicating the leading position of the United States in the field of myopia research. Hopefully, in the future, scholars around the world will endeavor to improve the quality of their research and publishing to expand their academic impact. Since these studies reflect regional contributions to myopia research and inter-regional academic collaborations, it is expected to provide useful information for national health policymakers and researchers in related fields. In addition, due to different ethnic groups and national policies, the prevention and management of myopia are different from country to country. In the future, multi-regional communications and collaborations are necessary.”
Additional comments:
- An explanation should be added as to what distinguishes this manuscript from others.
Response: Thank you for pointing this out. Similar to what we emphasized in the Conclusion section, the current study is the first bibliometric analysis focused on using multifocal lenses in myopic control. In addition, this study offers a comprehensive and objective analysis of the principal studies concerning the use of multifocal products for myopia control. Our findings highlight the research hotspots, key research directions, and the leading countries and regions over the past 24 years with the latest WOS database.
- Do the Authors plan further studies in this field?
Response: Thank you for pointing this out. It is really important for us to take a deep consideration about the future research direction. I think this bibliometric study has the potential to make a large, if not massive, positive contribution to the global literature, given the global increment myopic prevalence as well as the related health and economic burden. Given that numerous studies have demonstrated the efficacy of multifocal lenses in myopia control, our future research will focus on the mechanisms of multifocal lenses in myopia control, including potential theories such as accommodation and retinal contrast sensitivity. This is our plan for the further studies.

Reviewer 2 Report
Comments and Suggestions for Authors
I congratulate the authors on their detailed report about the bibliometric analysis of the studies on multifocal lenses for myopia control. I have added a few comments to the main text regarding the sections of the study that require revision. I would be happy to reevaluate the study after these revisions have been made.
line 66: "R programming environment" - Can you provide the full information -not only version- about the softwares (inluding the developer and the year) mentioned in this section
line 155-157: MCL may have a center-near or center-distance design as well as EDOF design. While covering MFLs in this sentence, the authors mentioned only center-distance lenses, which should be corrected.
line 162: Correct to "center", as you use "center" in the following sentences.
line 163-165: What is the purpose of this sentence? Also, it that information belong to a specific patient? Can you please revise this sentence
line 184: "Spherical" equivalent (SE) - please write with lowercase letter
line 185: "SVCL" Is it the same with SVL? If so, use only one of these abbreviations
line 187: Please add WMD in the first place you use "weighted mean difference", then you can use this abbreviation
line 225: You may use VA instead
line 232: Do you mean "+2.00 Add", if so please clarify.

Author Response
Dear reviewer 2,
Thank you so much for giving us this opportunity to revise our manuscript, which was significantly improved with your constructive suggestions. The corresponding revisions highlighted in the re-submitted files, and the point-to-point responses to the reviewer’s comments are listed below:
Comments: I congratulate the authors on their detailed report about the bibliometric analysis of the studies on multifocal lenses for myopia control. I have added a few comments to the main text regarding the sections of the study that require revision. I would be happy to reevaluate the study after these revisions have been made.
- line 66: "R programming environment" - Can you provide the full information -not only version- about the softwares (including the developer and the year) mentioned in this section
Response: Thank you for your suggestion, we provided the full information including the developer and the year about all the software mentioned in this section, at line 69, 70, 73 and 77.
- line 155-157: MCL may have a center-near or center-distance design as well as EDOF design. While covering MFLs in this sentence, the authors mentioned only center-distance lenses, which should be corrected.
Response: Thank you for your suggestion. Actually, the multifocal contact lens we used in myopia control typically incorporates a central zone to provide clear distance vision with relatively more positive peripheral lens power, which either increased gradually towards the periphery (progressive design) or presented as discrete peripheral annular zones (concentric ring design). This kind of lens design is based on the peripheral defocus theory of myopia, which posits that inducing peripheral myopic defocus can facilitate myopia control. To date no studies have published myopia control efficacy using center-near multifocal contact lenses. We corrected this sentence in the document at line 163, which is “The design of MCLs for myopia control typically features a central zone dedicated to distance correction, surrounded by concentric peripheral zones that incorporate a relative addition power of +2.00 Diopters (D) or greater”.
- line 162: Correct to "center", as you use "center" in the following sentences.
Response: Thank you, we corrected it in the manuscript.
- line 163-165: What is the purpose of this sentence? Also, it that information belongs to a specific patient? Can you please revise this sentence?
Response: Thank you for your suggestion. In fact, this sentence is based on the corresponding reference, which means MCLs could induce varying levels of peripheral myopic defocus on the retina. In addition, the degree of peripheral defocus induced by multifocal lenses needs to be considered when making myopia control strategies. So this sentence was revised as “MCLs are designed to create varying levels of peripheral myopic defocus on the retina.”
- line 184: "Spherical" equivalent (SE) - please write with lowercase letter
Response: Thank you, we corrected it in the manuscript.
- line 185: "SVCL" Is it the same with SVL? If so, use only one of these abbreviations
Response: Thank you, SVCL means single-vision contact lenses as we mentioned at line 206, while SVL means single-vision lenses as we mentioned at line 204, including spectacles and contact lenses. Since “single-vision contact lenses” occurred only once in the manuscript, we deleted the abbreviation.
- line 187: Please add WMD in the first place you use "weighted mean difference", then you can use this abbreviation
Response: Thank you for your advice, since “weighted mean difference” occurred only once in the manuscript, we used it instead of the abbreviation.
- line 225: You may use VA instead
Response: Thank you, we corrected it in the manuscript.
- line 232: Do you mean "+2.00 Add", if so please clarify.
Response: Thank you for your advice. It is "+2.00 Add”, so we revised the sentence: “But in a double-masked study, contrast sensitivity function was reduced specifically with +2.0D add power MCLs, not with +4.0D add power lenses”. We also added the corresponding reference.

Reviewer 3 Report
Comments and Suggestions for Authors
This manuscript gives a thorough review of the research works during the last 24 years about multifocal lens, including both multifocal contact lenses and multifocal spectacle lenses, for mitigating myopic progression. Although focusing on a very specific topic in ophthalmology, it covers quite a broad range of clinical optics and gives a major development trend in this field. Being an researcher in display and visual technology, I highly recommend this manuscript for Healthcare. Some minor comments are given as follows for clarifying some details in the data organization and categorization.
1. In Fig. 3 for the number of publications by country or region, there are quite a number of joint efforts across different countries and regions. How those publications from joint efforts were counted? Would they be double counted, or only counted for the country or region with the first author? With the total number of 234 research articles, they should have been double counted or even multiple counted, depending on the number of countries or regions involved. Would some hidden information be revealed with differentiating independent publication and joint publication?
2. In both Fig. 4 and Fig. 5A, there are connection line with hierarchy at the left most of the diagram. What is the meaning of these connection lines?
Author Response
Dear reviewer 3,
Thank you so much for giving us this opportunity to revise our manuscript, which was significantly improved with your constructive suggestions. The corresponding revisions highlighted in the re-submitted files, and the point-to-point responses to the reviewer’s comments are listed below:
Comments: This manuscript gives a thorough review of the research works during the last 24 years about multifocal lens, including both multifocal contact lenses and multifocal spectacle lenses, for mitigating myopic progression. Although focusing on a very specific topic in ophthalmology, it covers quite a broad range of clinical optics and gives a major development trend in this field. Being an researcher in display and visual technology, I highly recommend this manuscript for Healthcare. Some minor comments are given as follows for clarifying some details in the data organization and categorization.
- In Fig. 3 for the number of publications by country or region, there are quite a number of joint efforts across different countries and regions. How those publications from joint efforts were counted? Would they be double counted, or only counted for the country or region with the first author? With the total number of 234 research articles, they should have been double counted or even multiple counted, depending on the number of countries or regions involved. Would some hidden information be revealed with differentiating independent publication and joint publication?
Response: Thank you very much for pointing out this issue. When utilizing the Bibliometrix package for bibliometric analysis, the national affiliation of a paper is commonly determined based on the country information of the corresponding author. This is a standard approach as the corresponding author often represents the primary research hub and the location from where the research was coordinated and communicated. The bibliometrix package can extract this information from the metadata of the publications in databases like Web of Science.
We must admit that in the previous data analysis, we used the nationality of all authors to analyze the countries of the article, so 47 countries were included. However, after reviewing the previous literature and discussing with all authors, we think it is more reasonable to use the nationality of corresponding authors for analysis, so 31 countries were included in the final analysis. We corrected it in the results section (line 95). Thanks again for helping us thinking deeply into the data we obtained.
In addition, as you suggested, we reanalyzed the data, and differed the independent publication (in this article, it’s known as single country publication, SCP) and joint publication (in this article, it’s known as multiple country publication, MCP). The results was like this: the United States led the publication count with SCP of 51 articles and MCP of 11 articles, followed by China with SCP of 20 articles and MCP of 10 articles, Australia with SCP of 21 articles and MCP of 9 articles, Spain with SCP of 10 articles and MCP of 8 articles. Also, the United States has the highest proportion (82.3%) of independent publications among countries with more than five publications. The proportion of MCP for most countries ranges from 30-50%, including Australia, China, Spain and Germany. The detailed result of this part can be found in the results section (line 96-107) and Fig. 3A.
Fig. 3B is a chord chart illustrating international research collaboration. It includes only countries with clear collaborative ties, focusing on significant partnerships. Numbers on the chart indicate the count of co-authored papers between countries during the analyzed period, with larger figures reflecting more frequent collaborations. In cases where the authors of a paper come from multiple countries, it is common to consider each pair of partnerships between those countries as a collaboration. For example, if the author of the paper is from countries A, B, and C, then the collaboration between countries A and B, A and C, and B and C will be counted as one in the chord diagram. This counting method helps to more accurately reflect the strength of cooperation and network structure between countries.
- In both Fig. 4 and Fig. 5A, there are connection line with hierarchy at the left most of the diagram. What is the meaning of these connection lines?
Response: Thank you very much for pointing out this issue, which is really necessary to be clarified in the figure legends to make readers get a better understanding of our analysis.
The heatmap's dendrogram on the left delineates the outcomes of hierarchical clustering, a method that structures data points into a hierarchical tree, illustrating the aggregation of keywords into clusters based on their similarity. The vertical lines signify the merging points of two keywords or clusters, with line height reflecting the dissimilarity at merger, indicating greater dissimilarity with increased distance. Horizontal lines map the keywords' positions within the tree, each anchoring to a node. The tree's branching architecture uncovers the hierarchical interrelations among keywords, from individual terms to the comprehensive cluster at the summit. This visualization is pivotal in scientific literature, delineating the convergence of themes within a research domain and their temporal evolution. This concise visual summary encapsulates the research trends and thematic interconnections, providing a foundational basis for further scholarly inquiry.
We have also added a description of the connection lines in figure legends for Fig. 4 and Fig. 5A, as we listed below:
Figure 4. Number of publications by journal, with the sidebar using a color-coded system to denote the corresponding counts. The left dendrogram on this heat map illustrates hierarchical clustering outcomes, organizing journals into a tree structure based on similarity. Vertical lines mark journal cluster merges, with height indicating dissimilarity; greater heights suggest higher dissimilarity. Horizontal lines show journal positions, each linking to a tree node. The tree structure reveals journal hierarchical relationships, from individual terms to the overall cluster.
Figure 5. A, Annual frequency of keywords, with the sidebar using a color-coded system to denote the corresponding counts. The left dendrogram on this heat map illustrates hierarchical clustering outcomes, organizing keywords into a tree structure based on similarity. Vertical lines mark keyword cluster merges, with height indicating dissimilarity; greater heights suggest higher dissimilarity. Horizontal lines show keyword positions, each linking to a tree node. The tree structure reveals keyword hierarchical relationships, from individual terms to the overall cluster.

Reviewer 4 Report
Comments and Suggestions for Authors
Firstly, I'd like to congratulate the authors on their analysis, which attempted to track scientific advances in multifocal lenses for myopia management in the twenty-first century. It was stated that this bibliometric study provides a ground-breaking overview of research trends and evolution in the use of multifocal lenses for myopia management. Furthermore, these data provide a thorough understanding of current research frontiers and hot topics, serving as a significant resource for prospective research.
Given that myopia is a top public health priority, the increase in its prevalence has become a serious public health concern. By 2050, there are expected to be about 1 billion high myopes globally. The time to act is now. Practitioners should see myopia as a genetic and environmental disorder with distinct visual and economic consequences. Matter of fact, the rise in global urbanization, which appears to encourage myopia development, along with the rise in the economic burden associated with myopia, makes addressing myopia a top public health concern.
In this context, beyond behavioral therapies, as stated by the authors in their study, a variety of therapy modalities, including optical and pharmaceutical approaches, have been investigated to reduce myopic development. Multifocal lenses are an optical intervention method that includes both multifocal contact lenses (MCLs) and multifocal spectacle lenses, which have long been employed to slow myopia development. These revolutionary lens designs were created to cause a peripheral defocus on the retina. This seeks to diminish peripheral hyperopic defocus and hence reduce a crucial stimulant for axial elongation.
Overall, the authors highlighted a topic that has been thoroughly explored around the world, albeit from a distinct perspective. I feel this bibliometric study has the potential to make a large, if not massive, positive contribution to the global literature, given the global increment myopic prevalence as well as the related health and economic burden.
Minor issue
· Flow chart of scientometric analysis:
o A bibliometric analysis originally included 259 documents; however, if I'm not mistaken, only 234 articles (196) and reviews (38) research were analyzed. Would the authors adequately address this issue to ensure a relatively clear understanding and the overall integrity of the study?
Author Response
Dear reviewer 4,
Thank you so much for giving us this opportunity to revise our manuscript, which was significantly improved with your constructive suggestions. The corresponding revisions highlighted in the re-submitted files, and the point-to-point responses to the reviewer’s comments are listed below:
Comments:
Minor issue
- Flow chart of scientometric analysis:
o A bibliometric analysis originally included 259 documents; however, if I'm not mistaken, only 234 articles (196) and reviews (38) research were analyzed. Would the authors adequately address this issue to ensure a relatively clear understanding and the overall integrity of the study?
Response: First of all, thank you very much for your recognition of our manuscript. Secondly, thanks again for pointing out this issue which is very important to be clarified in the understanding of our analysis. This bibliometric analysis originally included 259 documents, with 25 documents excluded since these documents were meeting abstracts, proceeding papers, early accesses, editorial materials and letters, which did not meet our inclusion criteria. Figure 1 was revised to show this (red box).

Round 2
Reviewer 1 Report
Comments and Suggestions for Authors
The manuscript has been revised sufficiently
Reviewer 2 Report
Comments and Suggestions for Authors
This version of the manuscript is suitable for publication.
Reviewer 3 Report
Comments and Suggestions for Authors
The authors have made complete revision to respond to the issues raised in the previous review. This manuscript is recommended for the publication in Healthcare.
Reviewer 4 Report
Comments and Suggestions for Authors
Thanks to the authors for an excellent revision. I honestly believe that the revision process improved the work enough to be published in its current version in the journal Healthcare.